# Clinical Application of the Food Compass Score: Positive Association to Mediterranean Diet Score, Health Star Rating System and an Early Eating Pattern in University Students

**DOI:** 10.3390/diseases10030043

**Published:** 2022-07-07

**Authors:** Paraskevi Detopoulou, Dimitra Syka, Konstantina Koumi, Vasileios Dedes, Konstantinos Tzirogiannis, Georgios I. Panoutsopoulos

**Affiliations:** 1Department of Nutritional Science and Dietetics, Faculty of Health Sciences, University of Peloponnese, New Building, Antikalamos, 24100 Kalamata, Greece; viviandeto@gmail.com (P.D.); beta1967@hotmail.com (D.S.); vdedes@hotmail.com (V.D.); 2Department of Clinical Nutrition, General Hospital Korgialenio Benakio, Athanassaki 2, 11526 Athens, Greece; linakoumi@hotmail.com; 3Internal Medicine Department, Mediterraneo Hospital, 16675 Athens, Greece; tzirocon@gmail.com

**Keywords:** food compass score, health star rating, Mediterranean diet, breakfast, meal patterns

## Abstract

Nutrient profiling systems (NPS) assist consumers in food choices. Several scores based on NPS have been proposed, but data on their clinical application are lacking. The food compass score (FCS) is a newly developed NPS per 100 kcal (from 1 “least healthy” to 100 “most healthy”). We examined the correlations of FCS with other indices, food groups, and meal patterns. A total of 346 students of the University of the Peloponnese (269 women and 77 men) participated. Dietary habits were evaluated with a food frequency questionnaire, and FCS, health star rating score (HSR), and MedDietScore were computed. Meal and snack frequency consumption was reported. Principal component analysis revealed three meal patterns: “early eater” (breakfast, morning snack and afternoon snack), “medium eater” (lunch and dinner), and “late eater” (bedtime snack). Pearson partial correlations between ranked variables were used to test the correlation coefficients between FCS, other scores, and meal patterns, after adjustment for age, sex, BMI, and underreporting. FCS was positively correlated to HSR (rho = 0.761, *p* ≤ 0.001) in a multi-adjusted analysis. In the highest tertile of MedDietScore FCS was also positively correlated to MedDietScore (rho = 0.379, *p* < 0.001). The FCS was positively correlated with juices, high-fat dairy, vegetables, legumes, fruits, and olive oil and negatively correlated with sodas, alcoholic drinks, red meat, refined grains, sweets, fats other than olive oil, fast foods, and coffee. In addition, it related positively to the “early eater” pattern (rho = 0.207, *p* < 0.001). The FCS was associated with other quality indices and better nutritional habits, such as being an early eater.

## 1. Introduction

Front-of-pack (FOP) nutrition labels score food quality based on nutrient profiling systems (NPS) using specific criteria and intend to assist consumers in following a healthier diet [1]. Several individual dietary scores, based on NPS, have been proposed, such as “British Food Standards Agency” (FSA) in the UK, “Health Star Ratings” (HSR) in Australia and New Zealand, “Nutri-Score” in France, Germany, Belgium, the Netherlands, and Spain, etc. [1]. Most of them focus on “standard” nutrients or food groups with adverse (such as energy, saturated fat, sodium, and sugar) or beneficial effects (such as fiber, fruit, and vegetables). The majority of the various NPS models evaluated by the World Health Organization [2] had not undergone any type of validation. Meanwhile, data on NPS application regarding chronic diseases and mortality are only recently being published and mainly come from Spanish and French populations [3,4,5,6,7,8,9,10], while there are limited data on the relation of NPS scores with meal patterns [11]. However, even for the most widespread scores, such as the Nutri-score, gaps have been identified and adaptations have been proposed ad hoc, such as the incorporation of dietary fiber [12] or the categorization of olive oil from “C-yellow” (original Nutri-Score category) to “A-dark green” (FSAm NPS value) [10]. Similarly, in a recent government review of the HSR index it was observed that breakfast cereals, bars, sweetened milks, and confectionary were over-scored, while healthy oils were underscored, which led to the creation of an updated version in 2020 [13].

The newly developed food compass score (FCS) is a promising NPS that addresses such inconsistencies by including cutting-edge science. For its creation 54 attributes were considered in the following domains: nutrient ratios, vitamins, minerals, food ingredients, additives, processing, specific lipids, fiber, protein, and phytochemicals, ensuring a “holistic”, expanded assessment of each food profile [14]. For example, in view of the scientific evidence, the FCS does not score total fat or saturated fat, as most other NPS do, but unsaturated:saturated fat ratio, while it incorporates carbohydrate quality (carbohydrate:fiber ratio) and mineral quality (potassium: sodium ratio) [14]. Moreover, it scores positively the presence of phytonutrients, specific nutrients previously under-represented in NPS (such as vitamin D and choline), and specific fatty acids (such as omega-3 and medium-chain fatty acids), while it scores negatively other food components, such as additives and ultra-processed foods, based on the NOVA score [14].

On the antipode of NPS derived scores stands the Mediterranean diet, which has been extensively studied for its beneficial effects [15]. Indeed, the use of various scores for the assessment of Mediterranean Diet adherence, such as the MedDietScore, has been associated with healthy aging [16] and protection against cardiovascular and other diseases [17,18]. To our knowledge, only one study has assessed the relation of nutritional quality as assessed by the NPS to Mediterranean diet adherence [11]. It is thus plausible to investigate the relation of a novel NPS, such as the FCS, with this protective dietary pattern, especially when it comes to a Mediterranean population.

Thus, the aim of the present study was to apply the newly derived FCS and examine its correlations with other indices, namely the HSR and the MedDietScore, food groups, and meal patterns in Greek students enrolled at the University of the Peloponnese.

## 2. Materials and Methods

A cross sectional survey was performed on students enrolled at the University of the Peloponnese. More particularly, students from the University of Peloponnese (“Department of Nursing”, “Department of Sports Organization and Management”, “Department of Philology”, “Department of History, Archaeology and Cultural Resources Management” and the Technical Educational Institution of Peloponnese (fields of Management, Agriculture and Food Technology) were invited. Since, in 2019, the Technical Educational Institution of Peloponnese was integrated with the University of Peloponnese, we refer to our sample as “students from the University of Peloponnese” for simplicity reasons.

It is noted that the University of the Peloponnese has several departments, in five different towns, i.e., Kalamata, Sparti, Tripoli, Korinthos, and Nafplio. We followed a “convenience sampling” technique, mainly due to geographical constraints, and students from three different towns participated (Kalamata, Sparti, and Tripoli). As far as the recruitment method is concerned, paper advertisements were placed at the university and e-mails were centrally sent to all students of the departments that were included. Moreover, volunteers that agreed to participate were advised to “share the news” and invite their colleagues to participate.

A total of 346 students participated in the study (269 women and 77 men), with an age range 18–47 years. The total students at the University at the time of measurements were 4004, which means that the sample of the present study constituted 8.6% of the University students. The students were enrolled in various semesters of their studies (i.e., they were not all “freshmen”). The study protocol was approved by the University’s Ethics Committee (Faculty of Human Movement and Quality of Life Sciences). All procedures were in accordance with the Declaration of Helsinki (1989) of the World Medical Association, as revised in 2013. Informed consent was given by the participants and the president of each department. 

### 2.1. Anthropometry

Weight was measured with a digital scale (Seca) to the nearest 0.1 kg, in light clothing. Height was measured with a stadiometer to the nearest 0.1 cm, without shoes. Body mass index (BMI) was then calculated as the ratio of weight (in kilograms) to height^2^ (in meters squared). The participants were then classified as underweight, normal weight, overweight, and obese according to the World Health Organization (WHO) criteria [19]. 

### 2.2. Nutrition Assessment

For the purposes of the present study, we developed a semi-quantitative food frequency questionnaire (FFQ), which included 156 foods and beverages and 9 possible frequencies, i.e., never, sometimes per year, once per month, 2–3 times per month, once per week, twice per week, 3–4 times per week, 5–6 times per week, and every day. The portions used in the questionnaire reflected ordinary portions consumed, i.e., 250 mL for milk, juices, and other liquids; 30 g for cheese/ham; 200 g for yogurt; ½ cup for boiled vegetables; 1 cup for raw vegetables; ½ cup for rice and cereals; 1 cup for legumes; one medium fruit; 85 g for meat; and 170 g for fish. The Cronbach’s alpha coefficient for internal consistency was 0.89. Food group consumption was estimated by grouping food items into the following food groups: juices, sodas, liquid calories, alcoholic drinks, low-fat dairy (including cheese), high-fat dairy (including cheese), total dairy (sum of low-fat and full-fat dairy), vegetables, legumes, eggs, red meat, poultry, fish and seafoods, refined grains, whole grains, honey and marmalade, fruits, sweets, other fats, olive oil, tea, coffee, nuts, and fast foods. 

For assessment of Mediterranean Diet adherence, MedDietScore was calculated [20]. It is noted that appropriate transformations in food portions were made to calculate the MedDietScore, when needed.

For the calculation of individual FCS and HSR, a dietary index was calculated using energy-weighted means with the following equations [21]: $$\mathrm{Dietary}\;\mathrm{index}=\frac{\sum_{i=1}^{n}FSiEi}{\sum_{i=1}^{n}Ei}$$
 Ei=fi∗energy per portion
where *i* denotes a food or beverage ingested by the participant, *FSi* the food or beverage score, *Ei* the mean daily energy intake from this food or beverage, *n* the number of different foods or beverages, and fi the daily portions of the food item or beverage.

Higher values of the dietary index reflect a higher overall diet quality. The FCS classifies foods and beverages from “least healthy” (score 1) to “most healthy” (score 100) per 100 kcal and incorporates nutrient ratios, vitamins, minerals, food ingredients, additives, processing, specific lipids, fiber, protein, and phytochemicals in its calculation [14]. The HSR is a continuous score with 10 categories, which are rated from half a star to five stars [13]. It assesses energy, saturated fat, sodium, and total sugars per 100 g or ml, while it considers certain “positive” points of a product for fruit, vegetable, nuts, legumes, dietary fiber, and protein content [13].

Participants were asked to report their weekly frequency of main meal and snack consumption. More particularly, for each meal or snack (breakfast, morning snack, lunch, afternoon snack, dinner, bedtime snack) the participants checked the frequency of consumption: <once per week, 1–2 times per week, 3–4 times per week, 5–6 times per week, or every day. Then, scores were assigned to each frequency category from 1 to 5, correspondingly, and the total score was calculated (meal score). 

The USDA food database was used to calculate energy intake, by attaching an energy value to each included food item [22]. Basal metabolic rate (BMR) was estimated from weight, age, and sex, using the Schofield equation for adults [23]. Underreporting was then assessed using energy intake/BMR ratio (<1.09 for women and <1.07 for men), as suggested when using an FFQ [24].

### 2.3. Statistical Analysis

Normality was tested with use of the Kolmogorov–Smirnoff test. Normally distributed continuous variables are presented as mean values ± standard deviation, while non-normal variables are presented as median and interquartile range. Categorical variables are presented as absolute numbers and relative frequencies (%). Since data were skewed, Spearman coefficients (unadjusted analysis) and Pearson partial correlation coefficients for ranked variables were used to test variable associations after adjustment for age, BMI, sex, and underreporting (adjusted analysis). For normally distributed raw or transformed variables the Pearson correlation coefficient was used. Sex comparisons for non-normal variables were made with the use of a Wilcoxon signed ranked test.

A principal components analysis (PCA) was applied to identify meal patterns. To decide the number of components to retain from the factor analysis, eigenvalues derived from a correlation matrix of the standardized variables were assessed. Components with an eigenvalue higher than 1 were kept for the data analyses. Moreover, a scree plot was used to confirm the previous decision. Three components of meal patterns were finally extracted. The Kaiser–Mayer–Oklin criterion and Bartlett’s test were used to evaluate a factor’s analysis performance. Since component scores are interpreted similarly to correlation coefficients, the meal patterns were specified in relation to scores of variables that were most associated with the component (absolute loading value >0.60). The information was rotated to increase the representation of each food to a component (Varimax rotation).

All reported *p* values are compared to a significance level of 5%. IBM SPSS Statistics for Windows version 22.0 (Armonk, NY, USA: IBM Corp.) software was used for all the statistical analyses. 

## 3. Results

### 3.1. Basic Characteristic of the Subjects

The basic characteristics of the subjects are shown on Table 1. Briefly, the participants were young (mean age, 19.6 y) and 44.2% were freshmen. The highest participation rate was observed for the Nursing department, followed by the Department of Philology; Department of History, Archaeology, and Cultural Resources Management; and the Department of Sports Organization and Management. More than 2/3 of the sample had normal weight and 16.8% were overweight. The prevalence of obesity was 2.9%. Moreover, a higher participation for women was observed (269 women vs. 77 men).

### 3.2. Food Group Intake of the Subjects

The food intake of the volunteers is shown on Table 2. The median daily intake of milk, fruits, and vegetables was 1.45, 0.9, and 2.4 portions correspondingly. The median weekly intake of legumes, fish, red meat, and poultry was 2.31, 0.51, 6.90, and 1.16 portions, respectively. As far as sex comparisons are concerned, it was noted that men had a higher full fat milk intake than women: 0.66 portions/day (0.18–1.39) vs. 0.45 portions/day (0.08–1.05), and higher red meat: 9.75 portions/week (6.19–15.78) vs. 6.18 portions/week (3.92–10.23), refined grains: 2.33 portions/day (1.44–3.51) vs. 1.97 portions/day (1.15–2.99), and egg intake: 1.98 item/week (1.07–3.86) vs. 1.16 item/week (0.46–1.99). Men also had a higher intake of nuts: 0.11 portions/day (0.02–0.28) vs. 0.03 portions/day (0.02–0.11) (*p* < 0.001) and higher consumption of sodas: 0.33 (0.06–0.68) vs. 0.17 (0.04–0.44) (*p* = 0.02) and alcohol: 0.30 (0.09–0.68) vs. 0.17 (0.04–0.51) compared to women (*p* = 0.02) (values are medians and interquartile range, data not shown on Table). Moreover, they had lower vegetables intake than women: 1.74 (1.06–3.50) vs. 2.51 (1.59–4.05 (*p* = 0.01) (values are medians and interquartile range, data not shown on table). It can be seen that almost 25% of the participants underreported their intake (54.5% of men and 16.4 of women, *p* < 0.05). 

### 3.3. Evaluation of Participant Diets Based on FCS, HSR, and MedDietScore

The median (interquartile range) of the scores computed was 47.6 (41.0–54.4) for the FCS, 3.12 (2.79–3.34) for the HSR, and 30.0 (27.0–33.0) for the MedDietScore. 

### 3.4. Relation of FCS, HSR, and MedDietScore with Age, Sex, and BMI

The FCS was positively correlated with age (Pearson rho = 0.176, *p* = 0.001, Spearman rho = 0.154, *p* = 0.004), while no correlation was found between age and HSR (Spearman rho = 0.07, *p* = 0.18) nor age and MedDietScore (Spearman rho = 0.06, *p* = 0.2).

The FCS was higher in women than men (median 48.6 vs. 43, *p* = 0.02). Women also had higher MedDietScores than men (*p* = 0.01), while no differentiation was evident for HSR (*p* = 0.1). There was no relation of FCS and HSR to BMI (partial rho = 0.033, *p* = 0.5 and partial rho = 0.011, *p* = 0.8, correspondingly) after adjustment for age, sex, and underreporting. MedDietScore was inversely related with BMI (partial rho = −0.148, *p* = 0.006), after adjustment for the same covariates (data not shown).

### 3.5. Correlation Coefficients between Food Groups and FCS, HSR, and MedDietScore

In Table 3, simple Spearman correlation coefficients, as well as partial correlation coefficients of ranked scores and ranked food group consumptions, are displayed, adjusted for age, sex, BMI, and underreporting. The FCS was positively correlated with HSR, juices, high-fat dairy, vegetables, legumes, fruits, and olive oil. The FCS was negatively correlated with sodas, alcoholic drinks, red meat, refined grains, sweets, fats other than olive oil, fast foods, and coffee. It is noted that, after splitting the sample in MedDietScore tertiles into participants with a high adherence to the Mediterranean diet (3rd tertile), the FCS and MedDietScore were positively correlated, after adjustment for age, sex, BMI, and underreporting (rho = 0.379, *p* < 0.001, *n* = 104) (data not shown). 

### 3.6. Comparative Classification of Volunteers in Tertiles for FCS, HSR, and MedDietScore

In Table 4, the classification of volunteers in the same tertile is displayed for FCS, HSR, and MedDietScore. As can be seen, only 15%, 5.2%, and 15% of the participants were classified to the first, second, and third tertile, correspondingly, for all three indices.

### 3.7. Identification of Meal Patterns and Association of Meal Patterns to FCS, HSR, and MedDietScore

Three meal patterns were identified from the PCA analysis, namely “early eater” (consuming breakfast, morning snack, and afternoon snack), “medium eater” (consuming lunch and dinner), and “late eater” (consuming bedtime snack), which explained 70% of the total variance **(**Appendix A). As far as the correlations with meal patterns are concerned, Pearson partial correlations with ranked variables were performed after adjustment for age, sex, BMI, and underreporting. The “early eater” pattern was positively associated with FCS, HSR, and MedDietScore (partial rho = 0.207, *p* < 0.001, partial rho = 0.178, *p* = 0.001, and partial rho = 0.186, *p* = 0.001, correspondingly). It is noted that the MedDietScore and the FCS (borderline significance) were also inversely related to the “late eater” pattern (rho = −0.125, *p* = 0.024 and rho = −0.104, *p* = 0.059, correspondingly) (Appendix A). 

## 4. Discussion

The present study describes, for the first time, the associations of a new developed score (FCS) with HSR, MedDietScore, food groups, and meal patterns in a Greek population, after adjusting for multiple confounders, such as age, sex, BMI, and underreporting.

The FCS was positively associated with other indices of food quality such as HSR and MedDietScore. The association of FCS with HSR has been previously demonstrated (Spearman rho = 0.67 in unadjusted analysis) [14], and this is in line with our results. Its association with the MedDietScore is presented for the first time, which further corroborates its significance. Similarly, in another study, lower FSA scores of individual meals/snacks were associated with lower scores regarding Mediterranean diet adherence [11]. It is noteworthy that the classification of participants in the same tertile of each index was relatively low. This may be explained by the inherent characteristics of the scores. For example, most white breads and white rice had HSR scores of 3.5–4.0 stars, but their FCS was less than 12 out of 100 [14], while other food groups such as legumes, nuts, and seeds scored highly for both scores [14]. In addition, the correlation coefficients of the two scores vary across various food groups, from 0.2 for vegetables to 0.7 for fruits [14]. Moreover, the HSR was designed for packaged foods, not foods cooked at home or mixed meals, which renders the comparisons between HSR and FCS or MedDietScore rather difficult. 

Regarding the FCS correlations with food groups, it was positively correlated with “health-promoting” food groups, such as fruits, vegetables, legumes, and olive oil, and negatively associated with “unhealthy” food groups, such as meat, sweets, fast foods, sodas, alcoholic drinks, refined grains, and fats other than olive oil. Similarly, FSA score has been positively related to fruit, vegetable, and fish consumption [21]. The positive correlation of FCS with juices and full-fat dairy, and its negative correlation with coffee, should be viewed critically, although the FCS was not related to fruit juices nor coffee and was negatively related to full-fat dairy in the unadjusted analysis. Owing to its design, the FCS rates 100% fruit juices higher than other NPS [14]. This may be an issue to reconsider in this newly developed index, although the role of 100% fruit juice in obesity is still debated [25]. Full-fat yogurt and cheese have lower scores than their low-fat alternatives (FCS scores for full fat yogurt: 81, low-fat yogurt: 87, full fat Swiss cheese: 40, low-fat Swiss cheese: 48) but the differences are not large. This may derive from the fact that high-fat dairy provides fat-soluble vitamins, such as vitamin D, which is a component of FCS. Not to mention that there is a scientific debate regarding full-fat dairy products, shifting towards not supporting a detrimental relationship between their consumption and cardiometabolic health [26]. For example, milk and yogurt contain polar lipids with anti-thrombotic activity [27]. Coffee is medium-to-high scored in the FCS (score for cappuccino coffee: 58, score for espresso coffee: 55, score for cappuccino coffee non-fat: 73). Its negative correlation with FCS is possibly due to the method of score calculation, i.e., score per 100 Kcal. As already noted by the score “inspirers”, some items such as coffee and tea require more study on their FCS scoring, since they have health effects but are low in calories [14].

The association of FCS with obesity was not significant. Data from the use of other indices have shown inverse relationships with obesity and metabolic syndrome [3,6], but the sample in those cohorts was larger than in the present study. Moreover, the scoring of FCS and FSA is different, since the first has a scale from 0–100 and the second a much narrower categoric scale, while it uses separate scoring algorithms for several food groups [14]. An interesting association of FCS and the other assessed scores was identified with the “early eater” pattern. MedDietScore has previously been positively associated with “eating breakfast” [28,29], while persons scoring high at another NPS (NRF9.3) had a higher intake of calcium and potassium at breakfast [30]. Indeed, the observed associations may originate from the incorporation of “healthy foods” at breakfast or morning snack (such as fruits and dairy) ensuring nutrient adequacy [31] or they may reflect a better lifestyle being embraced, with a generally healthy eating profile (i.e., higher scores of FCS, FSR, and MedDietScore). However, it should be noted that we have no data on the content/quality of breakfast and snacks consumed, while there is no consensus on their definition [24], so different results may occur if different assumptions are made.

The strengths of our study include the application of a brand-new index, the FCS score, which uses expanded evidence-based characteristics for assessing the quality of foods [14]. The FCS considers several food characteristics and scores all items uniformly using the same algorithm and cut-off points [14]. However, several limitations should be considered in the interpretation of our results. First, the cross-sectional design of our study could not reveal causal relationships between the investigated parameters. Second, the generalizability of the presented results may be limited, since our sample included university students from the area of Peloponnese. For example, the FCS should be used cautiously in low-income countries with high rates of malnutrition, since it is based per 100 Kcal and it is not intended to prevent vitamin and mineral deficiencies [32]. Our sample mostly included women, but to address this issue, sex-adjusted correlations are also presented. Moreover, several mistakes may have arisen during the estimation of dietary intake, but we have partially accounted for this by using underreporting as a confounding variable in the reported correlations. Indeed, since the FCS is calculated per 100 Kcal, the adjustment for underreporting is important. Other studies have also reported that the observed associations of other NPS may change after controlling for this confounding factor [11]. Finally, the FCS is based on US foods, and differences may exist concerning traditional Greek foods.

## 5. Conclusions

The present study represents the first application of the FCS score, since its launch in 2021. Its correlations with HSR and MedDietScore and food groups were plausible and it was connected to better nutritional habits, such as being an early eater. The “take-home message” is that our results point to the potential usefulness of this new dietary index, at least in young adults. Testing against health outcomes and application to larger populations could enhance the use of FCS to guide consumer behavior, as well as to shape food policy and new product development by the industry.

## Figures and Tables

**Table 1 diseases-10-00043-t001:** Descriptive characteristics of participants.

	Total	Valid %	Men	Valid %	Women	Valid %	*p*
			269		77		
**Age (years)**	19.61 ± 3.15		19.35 ± 1.97		19.68 ± 3.41		0.2
**Participants**	346 *(n)*						
**Year of enrollment**							
*1st*	153 *(n)*	44.2	39	50.6	114	42.3	0.2
*2nd*	56 *(n)*	16.2	23	29.8	33	12.2	<0.001
*3rd*	63 *(n)*	18.2	10	12.9	53	19.7	0.2
*4th*	72 *(n)*	20.8	4	5.19	68	25.2	<0.001
*missing*	2 *(n)*	0.6	1		1		
**Living area (before enrollment)**							
*<50,000 habitats*	79 *(n)*	28.6	16	36.7	63	29.2	0.2
*>50,000 habitats*	197 *(n)*	71.4	44	73.3	153	70.58	0.3
*missing*	70 *(n)*	20.2	17		53		
**Department**							
*Nursing*	120 *(n)*	34.7	22	28.5	98	36.4	0.2
*Philology*	88 *(n)*	25.4	19	24.6	52	19.3	0.3
*History, Archaeology and Cultural Resources Management*	71 *(n)*	20.5	17	22.0	71	26.3	0.4
*Sports Organization and Management*	15 *(n)*	(4.3	3	3.8	12	4.4	0.8
*Other*	52 *(n)*	15	16	20.7	36	13.3	0.1
**ΒΜΙ (kg/m^2^)**	22.0 (19.9–24.4)		21.6 (19.7–23.8)		23.7 (21.4–25.9)		<0.05
*Normal weight*	238 *(n)*	69	49 (*n*)	64.5	189 (*n*)	70.3	0.3
*Overweight*	58 *(n)*	16.8	23 (*n*)	30.3	35 (*n*)	13	<0.001
*Obese*	10 *(n)*	2.9	1 (*n*)	1.3	9 (*n*)	3.3	0.3
*Morbid obese*	2 *(n)*	0.6	0 (*n*)	0	2 (*n*)	0.7	0.4
*Missing*	1 *(n)*		1 (*n*)				

Data are presented as mean ± standard deviation for normally distributed variables or median (lower–upper quartile) (25–75th) for non-normally distributed variables. Categorical variables are displayed as frequencies (*n*) and valid %. BMI categories were defined according to the World Health Organization criteria [19]. BMI: body mass index. Student *t*-test, Mann–Whitney, or Chi–square test (for categorical variables) was used to compare means. *p* < 0.05 denotes statistical differences between men and women.

**Table 2 diseases-10-00043-t002:** Dietary intake, MedDietScore, FCS, and HSR score of the participants.

Portions per Day or Week	Median	Interquartile Range
**Full-fat dairy (milk, yogurt, and cheese) (per day)**	0.50	0.10–1.12
**Low-fat dairy (milk, yogurt, and cheese) (per day)**	0.75	0.24–1.45
**Total dairy (per day)**	1.46	0.90–2.21
**Fruits (per day)**	0.96	0.42–1.91
**Vegetables (per day)**	2.44	1.41–3.94
**Legumes (per week)**	2.31	1.14–3.81
**Fish and sea foods (per week)**	0.51	1.04–2.03
**Red meat (per week)**	6.90	4.10–11.30
**Poultry (per week)**	1.16	0.58–1.99
**Whole-wheat grains and products (per day)**	0.08	0.31–0.86
**Refined grains and products (per day)**	1.22	2.03–3.08
**Eggs (per day)**	0.58	1.16–2.19
**Nuts (per day)**	0.02	0.04–0.16
**Sweets (per day)**	1.78	0.9–3.13
**Olive oil (per day)**	1	0.5–1
**Fats other than olive oil (per day)**	0.24	0.06–0.66
**Fast foods (per day)**	0.08	0.03–0.16
**Juices (per day)**	0.6	0.26–1.08
**Sodas (per day)**	0.18	0.05–0.51
**Alcoholic drinks (per day)**	0.19	0.04–0.56
**Energy/BMR**	1.56	1.09–2.19
**Underreporting (%)**	24.9	
**MedDietScore**	30.0	27.0–33.0
**FCS**	47.6	41.0–54.4
**HSR**	3.12	2.79–3.34

Data are presented as mean ± standard deviation for normally distributed variables or median (lower–upper quartile) (25–75th) for non-normally distributed variables. Categorical variables are displayed as valid %. FCS: food compass score; HSR: health star rating; MedDietScore: Mediterranean diet score.

**Table 3 diseases-10-00043-t003:** (**a**) Spearman correlation coefficients of raw variables and (**b**) Pearson partial correlation coefficients of ranked variables adjusted for age, sex, BMI, and underreporting.

(a)		FCS	HSR	MedDietScore
**FCS**	rho (*p*)	-	0.794 (*p* < 0.001)	0.434 (*p* < 0.001)
**HSR**	rho (*p*)	0.794 (*p* < 0.001)	-	0.466 (*p* < 0.001)
**MedDietScore**	rho (*p*)	0.434 (*p* < 0.001)	0.466 (*p* < 0.001)	
**Juices**	rho (*p*)	ns	ns	0.131 (*p* = 0.015)
**Sodas**	rho (*p*)	−0.477 (*p* < 0.001)	−0.427 (*p* < 0.001)	−0.224 (*p* < 0.001)
**Alcoholic drinks**	rho (*p*)	−0.232 (*p* < 0.001)	−0.295 (*p* < 0.001)	ns
**Low-fat dairy**	rho (*p*)	0.128 (*p* = 0.01)	0.215 (*p* < 0.001)	0.267 (*p* < 0.001)
**High-fat dairy**	rho (*p*)	−0.115 (*p* = 0.03)	−0.233 (*p* < 0.001)	−0.333 (*p* < 0.001)
**Vegetables**	rho (*p*)	0.401 (*p* < 0.001)	0.355 (*p* < 0.001)	0.465 (*p* < 0.001)
**Legumes**	rho (*p*)	0.359 (*p* < 0.001)	0.357 (*p* < 0.001)	0.526 (*p* < 0.001)
**Eggs**	rho (*p*)	ns	ns	0.137 (*p* = 0.01)
**Red meat**	rho (*p*)	−0.367 (*p* < 0.001)	−0.322 (*p* < 0.001)	−0.142 (*p* = 0.008)
**Poultry**	rho (*p*)	−0.106 (*p* = 0.05)	ns	ns
**Fish–seafoods**	rho (*p*)	0.140 (*p* = 0.009)	0.124 (*p* = 0.02)	0.433 (*p* < 0.001)
**Refined grains**	rho (*p*)	−0.328 (*p* < 0.001)	ns	0.165 (*p* = 0.002)
**Whole grains**	rho (*p*)	0.299 (*p* < 0.001)	0.266 (*p* < 0.001)	0.582 (*p* < 0.001)
**Fruits**	rho (*p*)	0.434 (*p* < 0.001)	0.389 (*p* < 0.001)	0.489 (*p* < 0.001)
**Sweets**	rho (*p*)	−0.553 (*p* < 0.001)	−0.586 (*p* < 0.001)	ns
**Fats other than olive oil**	rho (*p*)	−0.314 (*p* < 0.001)	−0.378 (*p* < 0.001)	ns
**Olive oil**	rho (*p*)	0.165 (*p* = 0.002)	ns	0.141 (*p* = 0.009)
**Tea**	rho (*p*)	0.225 (*p* < 0.001)	0.197 (*p* < 0.001)	0.361 (*p* < 0.001)
**Coffee**	rho (*p*)	ns	ns	ns
**Nuts**	rho (*p*)	ns	ns	0.130 (*p* = 0.01)
**Fast foods**	rho (*p*)	−0.432 (*p* < 0.001)	−0.420 (*p* < 0.001)	−0.166 (*p* = 0.002)
**(b)** *Ranked variables of scores/food groups*		**FCS**	**HSR**	**MedDietScore**
**FCS**	rho (*p*)	-	0.761 (*p* < 0.001)	ns
**HSR**	rho (*p*)	0.761 (*p* < 0.001)	-	ns
**MedDietScore**	rho (*p*)	ns	ns	-
**Juices**	rho (*p*)	0.121 (*p* = 0.02)	ns	ns
**Sodas**	rho (*p*)	−0.392 (*p* < 0.001)	−0.322 (*p* < 0.001)	−0.247 (*p* < 0.001)
**Alcoholic drinks**	rho (*p*)	−0.161 (*p* = 0.003)	−0.248 (*p* < 0.001)	ns
**Low-fat dairy**	rho (*p*)	ns	0.155 (*p* = 0.004)	0.236 (*p* < 0.001)
**High-fat dairy**	rho (*p*)	0.116 (*p* = 0.03)	ns	−0.380 (*p* < 0.001)
**Vegetables**	rho (*p*)	0.243 (*p* < 0.001)	0.218 (*p* < 0.001)	0.408 (*p* < 0.001)
**Legumes**	rho (*p*)	0.140 (*p* = 0.01)	0.156 (*p* = 0.004)	0.488 (*p* < 0.001)
**Eggs**	rho (*p*)	ns	ns	0.129 (*p* = 0.01)
**Red meat**	rho (*p*)	−0.281 (*p* < 0.001)	−0.237 (*p* < 0.001)	−0.194 (*p* < 0.001)
**Poultry**	rho (*p*)	ns	ns	ns
**Fish–seafoods**	rho (*p*)	ns	ns	0.398 (*p* < 0.001)
**Refined grains**	rho (*p*)	−0.369 (*p* < 0.001)	ns	0.106 (*p* = 0.05)
**Whole grains**	rho (*p*)	ns	ns	0.560 (*p* < 0.001)
**Fruits**	rho (*p*)	0.284 (*p* < 0.001)	0.250 (*p* < 0.001)	0.442 (*p* < 0.001)
**Sweets**	rho (*p*)	−0.492 (*p* < 0.001)	−0.505 (*p* < 0.001)	−0.185 (*p* < 0.001)
**Fats other than olive oil**	rho (*p*)	−0.256 (*p* < 0.001)	−0.335 (*p* < 0.001)	ns
**Olive oil**	rho (*p*)	0.144 (*p* = 0.008)	ns	ns
**Tea**	rho (*p*)	ns	ns	0.340 (*p* < 0.001)
**Coffee**	rho (*p*)	−0.144 (*p* < 0.001)	−0.191 (*p* < 0.001)	ns
**Nuts**	rho (*p*)	ns	ns	0.119 (*p* = 0.02)
**Fast foods**	rho (*p*)	−0.352 (*p* < 0.001)	−0.321 (*p* < 0.001)	−0.202 (*p* < 0.001)

Only significant correlations are shown. FCS: food compass score; HSR: health star rating; MedDietScore: Mediterranean diet score; ns: non-significant (*p* > 0.05).

**Table 4 diseases-10-00043-t004:** Classification of participants in the same tertile for FCS, HSR, and MedDietScore.

	Q1	Q2	Q3	Total
**Participants classified in the same** **tertile with all indices (n, %)**	55 (15%)	18 (5.2%)	52 (15%)	125 (35.2%)
**Participants classified in the same** **tertile with FCS and HSR (n, %)**	86 (24.9 %)	62 (17.9%)	85 (24.6%)	233 (67.4%)
**Participants classified in the same** **tertile with FCS and MedDietScore (n, %)**	66 (19.1%)	33 (9.5%)	62 (17.9%)	161 (46.5%)

FCS: food compass score; HSR: health star rating; MedDietScore: Mediterranean diet score.

## Data Availability

Data utilized in the paper can be made available from the corresponding author upon request.

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
