# Peer review of "Clinical Application of the Food Compass Score: Positive Association to Mediterranean Diet Score, Health Star Rating System and an Early Eating Pattern in University Students"

_diseases, 2022, doi:10.3390/diseases10030043_

Round 1

Reviewer 1 Report

This study was a survey about dietary habits among a group of college students, using Food compass score (FCS), Health Star Rating score (HSR) and MedDietScore. They concluded that “The FCS was associated with other quality indices and better nutritional habits, like being an early eater.”

The manuscript was well-done; however, I think the below point need to clarify:

The participants included a total of 346 students (269 women and 77 men) of the University of the Peloponnese. The authors have to show how many total students of the University of the Peloponnese. Then the proportions of participant rate could be estimated.

In addition, how did they recruited?

Author Response

The participants included a total of 346 students (269 women and 77 men) of the University of the Peloponnese. The authors have to show how many total students of the University of the Peloponnese. Then the proportions of participant rate could be estimated.

In addition, how did they recruited?

 We would like to thank the Reviewer for the comments.

The total students of the University at the time of measurements were 4,004, which means that the sample of the present study constituted 8.6% of the University students. It is noted, however, that the percentage of Nursing and Philology students was relatively high (participating Nursing students: 120 out of 376 enrolled students i.e., 31.9% and participating Philology students: 88 out of 369 enrolled students i.e., 23.8%).

It is noted that the University of the Peloponnese has several departments in five different towns i.e., Kalamata, Sparti, Tripoli, Korinthos and Nafplio. We followed a ”convenience sampling” technique mainly due to geographical constraints. More particularly, students from three different towns participated (Kalamata, Sparti and Tripoli). As far as the way of recruitment is concerned, paper advertisements were put at the university and e-mails were centrally sent to all students of the Departments that were included. Moreover, volunteers that agreed to participate were advised to “share the news” and invite their colleagues to participate. (Please see lines 77-93 and 95-97 of the revised manuscript).

Reviewer 2 Report

The paper “Clinical application of the Food Compass Score: Positive association to Mediterranean Diet Score, Health Star Rating System and an early eating pattern in students enrolled at the University of the Peloponnese” reports the main findings of a study investigating the dietary habits of a sample of 346 University students with a food frequency questionnaire and the Food Compass Score.

The study presents an interesting topic, however it requires major revisions to be considered for publication.

The main big issue in the manuscript is the section “Results” and the section “Conclusions”. The two sections require extensive revision, first of all because the section Conclusion should be very short and immediate and contain just a quick “take home message”. Also the section “Results” should be organized in a more harmonized way, because there is only a short presentation of results and all Tables stand alone in the section. I suggest you try to alternate the Table and the presentation of data.

Please, find below some specific comments:

Affiliation: it is not necessary to repeat the affiliation for the corresponding author.

Abstract: it is not necessary to add the words: “Introduction”, “Methods”, “Results”, “Conclusions”. The total amount of involved subjects is here 345, while in the manuscript it is 346. Please, make it consistent.

Materials and Methods

-          Line 82-87: this part should be clarified. As it was built, the two sentences first affirm that University students were enrolled, then both university students and students from the Technical Educational Institution of Peloponnese.

Results

-          Line 170-171: As you discuss results also based on differences men vs female, I suggest specifying the percentage of women and men involved in the study.

-          Table 1: results should be also presented according to subject sex (male vs female).

-          I suggest, where possible, to distinguish men and women, as sex is a factor allowing results distinction. Among study limitations you should include the higher percentage of women versus men.

Author Response

The main big issue in the manuscript is the section “Results” and the section “Conclusions”. The two sections require extensive revision, first of all because the section Conclusion should be very short and immediate and contain just a quick “take home message”. Also the section “Results” should be organized in a more harmonized way, because there is only a short presentation of results and all Tables stand alone in the section. I suggest you try to alternate the Table and the presentation of data.

Thank you for the comment. The Results section has now an enriched text with subheadings to ease the reader. Table 1 was updated to include the results for men and women separately. Table 2 is re-organized in order to easily detect correlations (“rho” and “p” are displayed in the same line) and now only significant correlations are shown for simplicity reasons.

Moreover, the “Conclusions” section is shorter and includes a “take home message” (lines 354- 357 of the revised manuscript).

Please, find below some specific comments:

Affiliation: it is not necessary to repeat the affiliation for the corresponding author.

Answer: The change was made.

Abstract: it is not necessary to add the words: “Introduction”, “Methods”, “Results”, “Conclusions”. The total amount of involved subjects is here 345, while in the manuscript it is 346. Please, make it consistent.

Answer: The changes was made.

Materials and Methods

-          Line 82-87: this part should be clarified. As it was built, the two sentences first affirm that University students were enrolled, then both university students and students from the Technical Educational Institution of Peloponnese.

The Technical Educational Institution of Peloponnese was integrated with the University of Peloponnese in 2019 after a governmental decision to upgrade technical institutions. The measurements were taken before that time and that is why the names of participating departments are also shown as they were at the time of the measurements. At the moment, the term “Technical Educational Institution of Peloponnese” is not valid, so we preferred using the term “University of Peloponnese”. A clarification has been added in the manuscript as follows:

“More particularly, students from the University of Peloponnese (“Department of Nursing”, “Department of Sports Organization and Management”, “Department of Philology”, “Department of History, Archaeology and Cultural Resources Management”  and the Technical Educational Institution of Peloponnese (fields of Management, Agriculture and Food Technology) were invited. Since in 2019 the Technical Educational Institution of Peloponnese was integrated with the University of Peloponnese, we refer to our sample as “students from the University of Peloponnese” for simplicity reasons.

It is noted that the University of the Peloponnese has several departments in five different towns i.e., Kalamata, Sparti, Tripoli, Korinthos and Nafplio. We followed a ”convenience sampling” technique mainly due to geographical constraints and students from three different towns participated (Kalamata, Sparti and Tripoli). As far as the way of recruitment is concerned, paper advertisements were put at the university and e-mails were centrally sent to all students of the Departments that were included. Moreover, volunteers that agreed to participate were advised to “share the news” and invite their colleagues to participate”

.Results

-          Line 170-171: As you discuss results also based on differences men vs female, I suggest specifying the percentage of women and men involved in the study.

The change was made.

-          Table 1: results should be also presented according to subject sex (male vs female).

The change was made.

-          I suggest, where possible, to distinguish men and women, as sex is a factor allowing results distinction. Among study limitations you should include the higher percentage of women versus men.

Thank you for the comment. The differences between the dietary intake of the participants (men vs women) are now shown in the text and Table 1 now includes results according to sex.

Please see lines 183 and 185-202 of the revised manuscript.

The limitation of including more women in our sample was added. It is noted that adjustments were made for sex in the presented correlations to address this issue.

Please see line 343 of the revised manuscript.

Reviewer 3 Report

Dear authors,

Thank you for the interesting article. I have suggested minor corrections regarding the title of the paper, the number of participants in the trial in the abstract section, and dividing conclusions in discussion and conclusions separately. All my suggestions are provided within the text.

All the best and stay safe,

Author Response

Thank you for the comments. The suggested changes were made.

Round 2

Reviewer 2 Report

Dear Authors,

thank you for amending the manuscript according to suggestions.

As regards lines 78-84, in my opinion you can also omit specifying the different Departmentsm if you wish. But if you wish to have them there, it is fine.